

# Association between statin use and risk of gallstone disease and cholecystectomy: a meta-analysis of 590,086 patients

Yu Chang[1,*], Hong-Min Lin[2,*], Kuan-Yu Chi[3], Wan-Ying Lin[4] and Tsung-Ching Chou[1]

[1] Department of Surgery, National Cheng Kung University Hospital, College of Medicine, National Cheng Kung University, Tainan, Taiwan

[2] Department of Family Medicine, Chi-Mei Medical Center, Tainan, Taiwan

[3] Department of Education, Center for Evidence-Based Medicine, Taipei Medical University Hospital, Taipei, Taiwan

[4] Department of Family Medicine, Taipei Medical University Hospital, Taipei, Taiwan

[*] These authors contributed equally to this work.

Corresponding author
Tsung-Ching Chou,
chou5182@gmail.com

## ABSTRACT

**Background**. Statins have been reported to reduce the risk of gallstone disease. However, the impacts of different durations of statin use on gallstone disease have not been clarified. The aim of this study is toperform a systematic review with meta-analysis to update and to elucidate the association between statin use and the risk of gallstone disease and cholecystectomy.

**Methods**. Medline, Embase and Cochrane Library were searched from the inception until August 2022 for relevant articles investigating the difference in the risk of gallstone disease between statin users and non-users (PROSPERO, ID: CRD42020182445). Meta-analyses were conducted using odds ratios (ORs) with corresponding 95% confidence intervals (CIs) to compare the risk of gallstone disease and cholecystectomy between statin user and nonusers.

**Results**. Eight studies enrolling 590,086 patients were included. Overall, the use of statins was associated with a marginally significant lower risk of gallstone disease than nonusers (OR, 0.91; 95% CI [0.82–1.00]). Further subgroup analysis showed that short-term users, medium-term users, and long-term users were associated with a significantly higher risk (OR, 1.18; 95% CI [1.11–1.25]), comparable risk (OR, 0.93; 95% CI [0.83–1.04]), and significantly lower risk of gallstone diseases (OR, 0.78; 95% CI [0.68–0.90]) respectively, compared to nonusers.

**Conclusions**. Patients with medium-term or long-term use of statins without discontinuation are at a lower risk of gallstone disease or cholecystectomy.

## INTRODUCTION

Gallstone is one of the most common indications for hospitalization in developed countries (*Beckingham, 2001*). The prevalence of gallstone in the western population is between 10–30% (*Marschall & C, 2007*). Regarding the treatment of symptomatic cholelithiasis,

cholecystectomy is an optimal management with its surgical volume reaching up to 70,000 in the United States in 2008 (*Beckingham, 2001*; *Lammert & Miquel, 2008*). Gallstones primarily fall into two compositions, cholesterol and pigment stones, with the former accounting for 80–90% of all cases (*Marschall & C, 2007*). Nowadays, 3-hydroxy-3-methylglutaryl coenzyme A (HMG-CoA) reductase inhibitors, namely statins, are widely used for the treatment and prevention of cardiovascular disease due to their well-established therapeutic effects on low-density lipoprotein cholesterol (LDL-C), which is one of the major risk factors for atherosclerosis and coronary heart disease (*Awad et al., 2017*). In addition to the effects on cardiovascular system, statins significantly decrease cholesterol biosynthesis, which may potentially reduce cholesterol concentration in the biliary system (*Kallien et al., 1999*). Although it has been proven in animal studies that statin may alter biliary lipid composition, thereby reducing the risk of gallstone development (*Davis, Wertin & Schriver, 2003*; *Abedin et al., 2002*), in human studies it differed widely in the outcomes. Some studies ascertained statins' efficacy of decreasing cholesterol concentration in bile or dissolving gallstones (*Porsch-Ozçürümez et al., 2001*; *Smith et al., 2000*; *Chapman et al., 1998*; *Wilson et al., 1994*), whereas others found little effect of statins on bile composition (*Miettinen et al., 1998*; *Smit et al., 1995*; *Sharma et al., 1997*). The latest meta-analysis (*Kan et al., 2015*) in 2015 concluded that statin users have lower risk of gallstone disease. However, considerable heterogeneity has remained unexplored, and the impacts of different durations and discontinuation of statin use on gallstone varied in the literature. For example, while *Erichsen et al. (2011)* reported lower risk of gallstone disease among long-term user, more recently, *Biétry et al. (2016)* reported comparable risk of gallstone disease among long-term user and nonusers. Moreover, the definition of statin current user or former users was not consistent across published studies. Hence, we sought to perform a meta-analysis to integrate the available and relevant literature and determine the association between statin use and risk of gallstone disease and cholecystectomy.

## METHODS

We conducted the present systematic review and meta-analysis following Cochrane Handbook for Systematic Reviews and Interventions (*Julian Higgins et al., 2019*) and reported the results based on the Preferred Reporting Items for Systematic Reviews and Meta-Analyses and Meta-analysis Of Observational Studies in Epidemiology guidelines (Supplemental Method 1 and Supplemental Method 2). Electronic databases of Medline, Embase, and Cochrane library were searched from the inception until August 2022, encompassing all languages. This study was registered on PROSPERO (ID: CRD42020182445). Before the registration, we completed the formal screening of search results against eligibility criteria, since PROSPERO accepts registration for reviews that have not started data extraction. Two investigators (Y.C and K.Y.C) independently conducted the search to determine relevant studies to be included, and any discrepancy was addressed by reaching a consensus or by consulting a senior reviewer (T.C.C). Search details are presented in Supplemental Method 3.

## Eligibility criteria

Articles with the following criteria were included: (1) Randomized controlled trials, prospective studies, retrospective studies and case-control studies; (2) studies involving human adults without history of gallstone disease or cholecystectomy as participants; (3) studies reporting clinical outcome as with/without diagnosis of gallstone disease or record of cholecystectomy. Studies not investigating the use of statins were excluded. In case of duplicate studies with an accumulated number of patients or increased durations of follow-up, only the most complete reports were included.

## Data extraction

Two investigators (Y.C and K.Y.C) independently extracted relevant information from tables or results of eligible articles. Extracted data included first author name, publication year, country where the study was conducted, follow-up durations, number of participants, sex, age range, underlying diseases, and method used to identify cases. Additionally, we obtained adjusted odds ratio (OR) and standard error (SE) from each study. For studies that reported outcomes by relative risk (RR), since the extracted RR and SE were adjusted, the formula (*Grant, 2014*) for the conversion between RR and OR may be inappropriate. Therefore, we regarded the adjusted RR as OR since most of the included studies have incidence of outcome less than 10% and all ORs converted from RRs were between 0.5 and 2.5, correction may not be desirable (*Zhang & Yu, 1998*).

## Quality assessment

Two investigators (Y.C and H.M.L) independently completed a critical appraisal of included literature by using the Risk Of Bias In Non-randomized Studies—of Interventions (ROBINS-I) tool (*Sterne et al., 2016*) for retrospective cohorts and National Heart, Lung, and Blood Institute tool for case-control studies. Any item on which assessors did not reach a consensus was addressed through discussion with third investigator (T.C.C).

## Statistical analysis

Meta-analysis was conducted using R software with the "metafor" package (*Viechtbauer, 2010*) (Supplemental Method 4). Regarding the risk of gallstone disease, we used the inverse variance (IV) method to pool the odds ratio obtained from each study regarding risk of gallstone disease among statin users and non-users. The restricted maximum likelihood (REML) (*Harville, 1977*) method was exploited as a heterogeneity estimator for conducting random-effects meta-analyses given that the between-trial variance was inevitable. $P$ values <0.05 were considered statistically significant. Heterogeneity was assessed using $I^2$ statistics proposed by Higgins and Thompson (*Higgins, 2003*), with estimated values of $I^2<25\%$, $25\%<I^2<50\%$, and $I^2 \geq 50\%$ indicating low, moderate, and high heterogeneity, respectively. We aimed to explore the statistical heterogeneity of our meta-analysis by performing subgroup analysis based on the durations of statin use (long-term, medium-term and short-term) and current or former statin users.

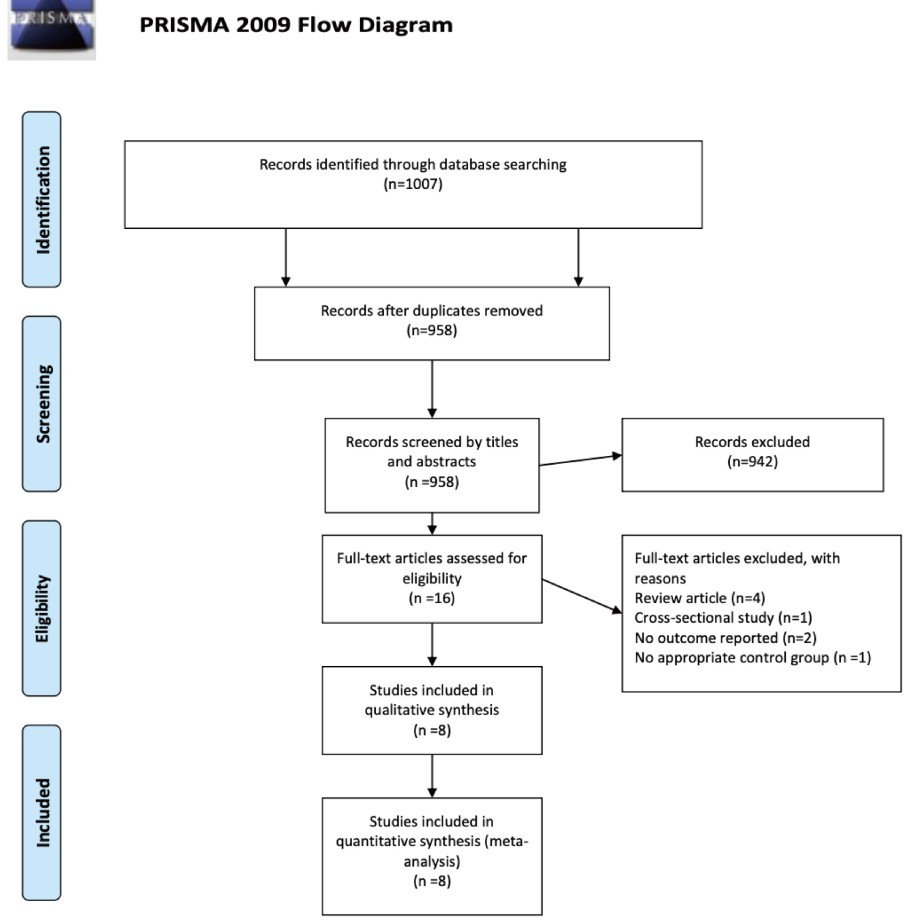

**PRISMA 2009 Flow Diagram**

**Figure 1  PRISMA flow diagram.** The PRISMA flow diagram demonstrates a total of 1007 potential references were extracted initially and meta-analysis included eight studies meeting the eligibility criteria.

## RESULTS

### Study selection

Our search strategy identified 1007 references from Medline, Embase, and Cochrane library. After title and abstract screening, we excluded duplicates ($n = 49$) and irrelevant references ($n = 942$) and retrieved 16 full texts for further review. Eventually, six case-control studies (*Erichsen et al., 2011*; *Biétry et al., 2016*; *Merzon et al., 2010*; *González-Pérez & García Rodríguez, 2007*; *Bodmer et al., 2009*; *Chiu et al., 2012*) and two retrospective cohort studies (*Tsai et al., 2009*; *Martin et al., 2016*) met the eligibility criteria for qualitative and quantitative synthesis. Figure 1 shows the flowchart of study selection.

### Characteristic of included studies and patients (Table 1)

Among eight studies (*Erichsen et al., 2011*; *Biétry et al., 2016*; *Tsai et al., 2009*; *Merzon et al., 2010*; *González-Pérez & García Rodríguez, 2007*; *Bodmer et al., 2009*; *Chiu et al., 2012*; *Martin et al., 2016*) enrolling 590,086 participants from 1994 to 2012, five studies (*Biétry*

**Table 1  Characteristics of included studies.**

| Study, year | Study type | Country | Inclusion period | Match method | Exclusion criteria | Case definition | Statin use |
|---|---|---|---|---|---|---|---|
| Biétry 2016 (*Harville, 1977*) | CC | Switzerland | 2008~2014 | 1:4 matched on age, sex, and index date | Cancer or HIV | Cholecystectomy only | Prior to index date †, Current user: last prescription <180 days Former user: last prescription >180 days |
| Bodmer 2009 (*Grant, 2014*) | CC | UK | 1994~2008 | 1:4 matched on age, sex, and index date | Alcoholism, drug abuse, cancer or HIV | First time diagnosed gallstone disease followed by cholecystectomy in 2 years; or cholecystectomy only | Prior to index date, Current user: last prescription <90 days Former user: last prescription >90 days |
| Chiu 2012 (*Sterne et al., 2016*) | CC | Taiwan | 1996~2009 | 1:1 matched on age, sex, and index date | Cancer | First time diagnosed gallstone disease followed by cholecystectomy in 2 years; or cholecystectomy only | Any prescription prior to index date |
| Erichsen 2010 (*Biétry et al., 2016*) | CC | Denmark | 1996~2008 | 1:10 matched on age, sex | Preexisting gallstone disease or liver, bile duct or pancreatic cancer | Diagnosis of gallstone, cholecystitis or a record gallbladder surgery (cholecystectomy or drainage) | Prior to index date, Current user: last prescription <90 days; Former user: last prescription >90 days |
| González-Pérez 2007 (*Julian Higgins et al., 2019*) | CC | UK | 1996~1996 | 1:4 matched on age and sex | Preexisting gallstone disease or cancer | Symptomatic gallstone | Prior to index date, Current user: last prescription <30 days; Former user: last prescription >30 days |
| Merzon 2010 (*Erichsen et al., 2011*) | CC | Israel | 2003~2006 | 1:4 matched on age and sex | N/A | Cholecystectomy due to gallstone disease | Prior to index date, Last prescription <180 days prior to index date |
| Martin 2015 (*Higgins, 2003*) | RC | US | 2003~2012 | 1:1 propensity matched | History of burn, trauma, statin use <90 days, or starting statin use after baseline period | Diagnosis of cholelithiasis | Statin use lasting over 90 days prior to index date |
| Tsai 2009 (*Kan et al., 2015*) | RC and PC | US | 1994~2000 | N/A | Prior cholecystectomy or gallstone disease or cancer | Self-reported cholecystectomy | Prior to index date, Current user: self-reported current use from inception to 2000, Former user: no present use in 2000 |

**Notes.**

CC, Case-control study; RC, retrospective cohort study; NA, not applicable.

*et al., 2016*; *Tsai et al., 2009*; *Merzon et al., 2010*; *Bodmer et al., 2009*; *Chiu et al., 2012*) recruited patients who underwent cholecystectomy due to gallstone diseases, and other three studies (*Erichsen et al., 2011*; *González-Pérez & García Rodríguez, 2007*; *Martin et al., 2016*) included patients with the diagnosis of gallstone diseases. Of note, five studies (*Erichsen et al., 2011*; *Biétry et al., 2016*; *Merzon et al., 2010*; *González-Pérez & García Rodríguez, 2007*; *Bodmer et al., 2009*) specifically categorized patients into statin current, past, or nonusers, and the other three studies (*Tsai et al., 2009*; *Chiu et al., 2012*; *Martin et al., 2016*) referred to statin users as any statin prescription prior to the index date, defined as the first-time diagnosis of gallstone diseases or the date of cholecystectomy in patients without the record of cholelithiasis. Detailed patient characteristics were summarized in Tables S1 and S2. Regarding the durations of statin use, three of the included studies (*Erichsen et al., 2011*; *Biétry et al., 2016*; *Bodmer et al., 2009*) classified the durations of statin use based on the numbers of prescription prior to the index date (Tables S3 and S4).

## Quality assessment of included studies

Table S5 demonstrated the summary of quality assessment for case-control studies using the National Heart, Lung, and Blood Institute tool, and the overall quality was good in three studies and the other three were fair. Table S6 demonstrated the risk of bias assessment using ROBINS-I tool for two retrospective cohort studies, which were evaluated as moderate risk of bias resulted from bias due to confounding.

## Risk of cholelithiasis and cholecystectomy

Overall, the pooled result of eight studies (*Erichsen et al., 2011*; *Biétry et al., 2016*; *Tsai et al., 2009*; *Merzon et al., 2010*; *González-Pérez & García Rodríguez, 2007*; *Bodmer et al., 2009*; *Chiu et al., 2012*; *Martin et al., 2016*) demonstrated that the use of statins was associated with a marginally significant lower risk of gallstone disease than nonusers, however, with substantial heterogeneity (OR, 0.91; 95% CI [0.82–1.00], $I^2 = 86\%$; Fig. 2).

## Durations of statin use

To explore the source of heterogeneity, we performed subgroup analysis according to the durations of statin use in three studies (*Erichsen et al., 2011*; *Biétry et al., 2016*; *Bodmer et al., 2009*) that exclusively classified the durations based on the number of prescriptions prior to the index date (short-term, 1-4; medium, 5-19; long-term, over 20). Notably, our subgroup analysis seemed to show a dose–response tendency (Fig. 3). Short-term use of statins is associated with higher risk of gallstone disease in both current (OR, 1.16; 95% CI [1.07–1.26], $I^2 = 0\%$; with gallstone disease: 804 patients; without gallstone disease: 4,430 patients) and former users (OR, 1.20; 95% CI [1.10–1.30], $I^2 = 35\%$; with gallstone disease: 717 patients; without gallstone disease: 3,469 patients). Medium-term current statin user had significant lower risk of gallstone disease compared to nonusers (OR, 0.86; 95% CI [0.81–0.91], $I^2 = 15\%$; with gallstone disease: 1,737 patients; without gallstone disease: 10,923 patients) and former users had comparable risk of gallstone disease compared to nonusers (OR, 1.05; 95% CI [0.91–1.21], $I^2 = 50\%$; with gallstone disease: 587 patients; without gallstone disease: 3,259 patients). Long-term use of statin significantly reduced the risk of gallstone disease in current users (OR, 0.73; 95% CI [0.62–0.87], $I^2 = 82\%$;

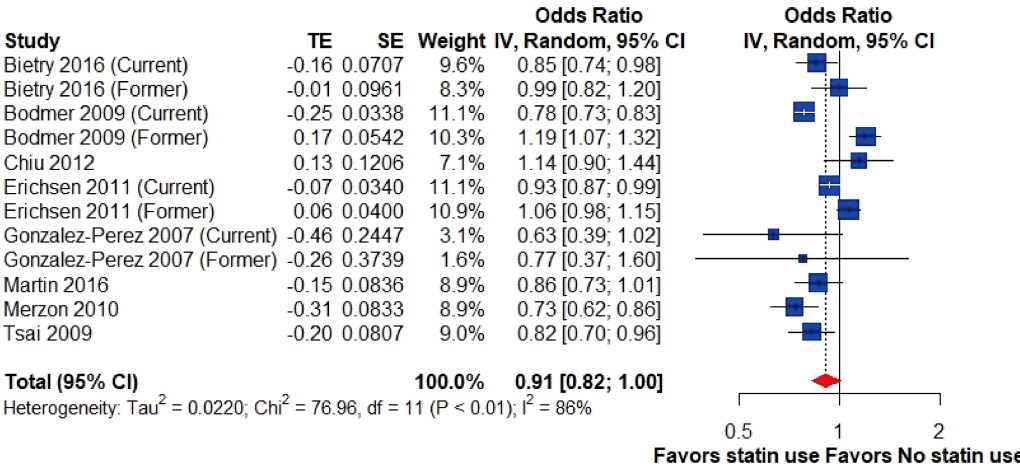

**Figure 2 Risk of gallstone disease among statin users and nonusers.** Forest plot for risk of gallstone disease among statin users *versus* nonusers. The size of squares is proportional to the weight of each study with the error bar showing the 95% CI. Red diamond represents the pooled estimate of OR with the 95% CI. CI, confidence interval; IV, inverse variance method; OR, odds ratio; SE, standard error of logOR; TE, logOR.

with gallstone disease: 1,451 patients; without gallstone disease: 9,096 patients) but former users had comparable risk of gallstone disease compared to nonusers (OR, 0.87; 95% CI [0.72–1.05], $I^2 = 0\%$; with gallstone disease: 204 patients; without gallstone disease: 1,327 patients).

Additionally, the pooled analysis of four studies (*Erichsen et al., 2011*; *Tsai et al., 2009*; *González-Pérez & García Rodríguez, 2007*; *Martin et al., 2016*) involving statin-users over 1-year period showed a significantly lower risk of gallstone diseases (OR, 0.83; 95% CI [0.77–0.89]; $I^2 = 4\%$; Fig. 4).

# DISCUSSION

## Risk of cholecystectomy and cholelithiasis

Statins are one of the most common prescribed regimens in the treatment and prevention of cardiovascular disease (*Chou et al., 2016*). In addition to lowering the risk of cardiovascular disease, some studies have demonstrated that statins may reduce the risk of urolithiasis (*Cohen et al., 2019*; *Sur et al., 2013*). Notably, our study revealed that statin users had marginally significant reduced risk of gallstone disease, compared to non-users, however, with high statistical heterogeneity ($I^2 = 71\%$).

One of the sources of heterogeneity would arise from the differences in demographics among included studies. For example, there were wide variation in prevalence of diabetes, ischemic heart diseases, and cerebrovascular diseases among included studies. Moreover, baseline characteristics that may be associated with gallstone disease such as hyperlipidemia, estrogen exposure or hormone therapy were only reported in two studies. On the other hand, we acquired adjusted ORs from included studies and pooled in meta-analysis,

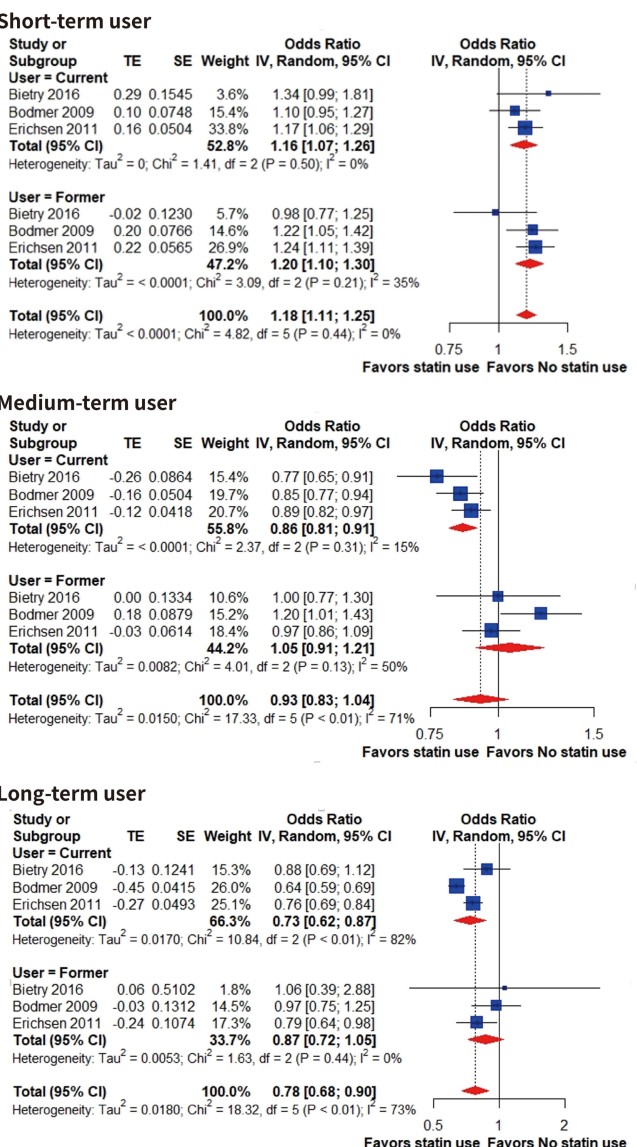

**Figure 3 Risk of gallstone disease based on different duration of statin use.** Forest plot for risk of gall-stone disease among short-term, medium-term and long-term statin users compared to nonusers. Sub-group analysis was performed based on current and former users. The size of squares is proportional to the weight of each study with the error bar showing the 95% CI. Red diamond represents the pooled estimate of OR with the 95% CI. CI, confidence interval; IV, inverse variance method; OR, odds ratio; SE, standard error of logOR; TE, logOR.

however, the variables adjusted for were not consistent in each study, which would contribute to the statistical heterogeneity.

Most of the included studies defined case as "cholecystectomy required". The use of statin may not only impact the formation of gallstone but the symptom severity of gallstone disease. For instance, *Suuronen et al. (1995)* found a decreased number of all cholecystectomies but increased number of laparoscopic cholecystectomy (LC) with statin
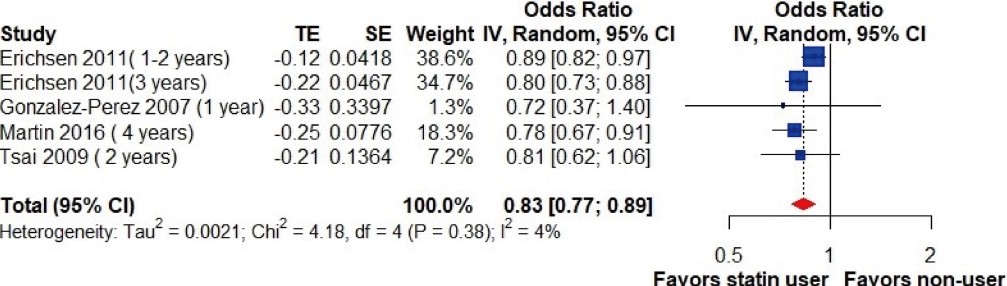

**Figure 4  Risk of gallstone disease among long-term statin users (over one year).** Forest plot for risk of gallstone disease among long-term statin users *versus* nonusers. Subgroup analysis was performed based on current and former users. The size of squares is proportional to the weight of each study with the error bar showing the 95% CI. Red diamond represents the pooled estimate of OR with the 95% CI. CI, confidence interval; IV, inverse variance method; OR, odds ratio; SE, standard error of logOR; TE, logOR.

use in their population-based study. They described one of the reasons may be an increased incidence of mild symptomatic gallstones leading to LC. The utilization of different approaches may reflect the severity of the disease, however, none of included studies investigated the difference between open and laparoscopic cholecystectomy.

## Gallstone disease risk of statin users with different durations of use

Our results demonstrated that there was a marginally significant reduction in the risk of developing gallstone disease among statin-ever users. The subgroup analysis showed a dose–response relationship between the number of prescription and the risk of gallstone diseases, and, of note, the protective effect of stain was only observed in medium and long-term current users. However, this result had to be interpreted with caution since the durations of statin use was solely based on the prescription of medication in medical records. Paucity of detailed dosage of statin prescription and information for drug compliance may hindered the accuracy of statin exposure. Moreover, the cut-off time selected for the definition of current and former users varied from 90 days to 180 days representing an obvious source of conceptual heterogeneity.

As a matter of fact, long-term statin users only accounted for 15% of all statin users in our analysis. We postulated that the underlying reason for such small proportion may be associated with either discontinuation of statins or low adherence to medications. Although statins are one of the most effective medications for the secondary prevention of cardiovascular diseases, non-compliance and discontinuation with this medication are common problems (*Navar et al., 2019*; *Toth et al., 2019*; *Wei et al., 2002*). Although patient-perceived side effect is the most common reason for statin discontinuation (*Bradley et al., 2019*), the incidence of statins' most common side effect, namely myopathy, is only estimated to be around 0.01% (*Thompson, Clarkson & Karas, 2003*). Of note, those not adhering to statins, mostly due to self-perceived stable disease condition (*Kamal-Bahl et al., 2007*), may experience relapsing dyslipidemia following statin discontinuation leading to worsening cardiovascular and survival outcomes (*Daskalopoulou et al., 2008*; *Rea et al., 2021*).

Consequently, we propose that the recurrence of dyslipidemia may be responsible for neutral and detrimental effects on gallstone formation in medium- and short-term users, respectively.

## Compare to previous synthesis

In our study, we included three more studies (*Biétry et al., 2016*; *González-Pérez & García Rodríguez, 2007*; *Martin et al., 2016*) compared with the previous one. In accordance with the study by *Kan et al. (2015)*, our results demonstrated patients with medium-term use and long-term use of statin users conferred to lower risk of gallstone disease. However, we addressed several concerns that were not well-elucidated in previous study. First, we clarified the impacts of statin discontinuation on gallstone disease. Second, we excluded cross-sectional study in our meta-analysis since this type of study is estimating the prevalence but incidence (*Sedgwick, 2014*). The cross-sectional study by *Caroli-Bosc et al. (2001)* in the previous meta-analysis might not contribute to statistical heterogeneity, however, conceptually, it brings between-study variance. Third, methodologically, we performed our meta-analysis through a more robust heterogeneity estimator based on the latest simulation study (*Langan et al., 2019*).

## Limitation

The findings of our meta-analysis must be interpreted within the context of the study limitations. First, we included case-control studies and retrospective cohort studies, which induced a potential bias due to the confounding that was generally not well-controlled. Second, we investigated the protective effect of statin on gallstone development; however, the definition of statin use, and gallstone disease were not consistent in each study. Third, baseline characteristics that may be associated with gallstone formation such as hyperlipidemia, estrogen exposure or hormone therapy were not reported in several studies, which hindered the accuracy of our findings. Last but not least, healthy user and adherer effect are two easily overlooked sources of bias, especially in observational studies (*Hrank, Patrick & Brookhart, 2011*). Patients who are willing to take and adhere to statins are more likely to be engaged with healthy behaviors, contributing to the protective effects of statins. Unfortunately, it is difficult to adjust these healthier behaviors through observational design, leading to unmeasured confounding bias.

# CONCLUSIONS

Based on our meta-analysis, long-term and medium-term use of statins without discontinuation lower the risk of gallstone disease or cholecystectomy. Nonetheless, given the observational nature of the included studies, we propose that further prospective studies should be undertaken to overcome the aforementioned limitations.

## Funding

The authors received no funding for this work.

### Competing Interests

The authors declare there are no competing interests.

### Author Contributions

- Yu Chang conceived and designed the experiments, performed the experiments, analyzed the data, prepared figures and/or tables, authored or reviewed drafts of the article, and approved the final draft.
- Hong-Min Lin performed the experiments, analyzed the data, prepared figures and/or tables, and approved the final draft.
- Kuan-Yu Chi conceived and designed the experiments, performed the experiments, prepared figures and/or tables, authored or reviewed drafts of the article, and approved the final draft.
- Wan-Ying Lin analyzed the data, authored or reviewed drafts of the article, and approved the final draft.
- Tsung-Ching Chou conceived and designed the experiments, authored or reviewed drafts of the article, and approved the final draft.

### Data Availability

This is a meta-analysis.

### Supplemental Information

Supplemental information for this article can be found online at http://dx.doi.org/10.7717/peerj.15149#supplemental-information.

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
