# Peer review of "Association between statin use and risk of gallstone disease and cholecystectomy: a meta-analysis of 590,086 patients"

_PeerJ, doi:10.7717/peerj.15149_

## Round 0.1 · original submission · Major Revisions

The findings are interesting and the authors should give due consideration to the comments of all three reviewers during a revised submission in order to improve the quality of the manuscript. Please clarify how the submitted findings are not redundant to those cited by reviewer #3. Further clarifications on the methods will be helpful.

Reviewer 1 ·

Basic reporting

The meta-analysis study of statins and risk of gallstone disease by Chang et al. investigates the relationship between statin use and the risk of gallstone disease. The authors analyzed 8 studies including 590,086 patients and calculated the odds ratio for 1) the overall group and for 2) subgroups divided by the duration of statin use. The authors found that long term statin users are at lower risk of developing gallstone disease.

The paper is very well structured and easy to follow. The introduction gave a clear description of the mechanism of action of statins as well as prior study which sufficiently motivates the present study. The results and discussion section were well-written, and the sub-headers helped the flow the descriptions.

Experimental design

The original research is within Aims and Scope of the Journal. The research question is well defined and the comparison with prior studies explained how the current study filled a research gap. This study is a rigorous meta analysis combining 8 previous studies with rigorous statistical analysis. The method section along with the query and the R script parameters in the supplementary material provide a good basis for reproducibility.

Validity of the findings

The findings are well supported by the results and data presented. Each statistical analysis was performed with case and controlled group with number of patients indicated. THe conclusions are well stated and clear.

Additional comments

I have the following comments and questions:
1. In the subgroup analysis, it will be informative to mention the number of patients in case vs. control. This information is in Supplementary Table 3 but was not actually referred to in text.
2. For the figures, please be more detail in explaining the figures. For example, in figure 2, please explain “TE”, “SE”, “IV”. In addition, for the Odds Ratio figure on the right, please describe what the red diamond is as well as the dotted line. What does the error bars stand for? It will also help clarify the figures if you add something to the effect “Each square represents the odds ratio calculated from a study in the table on the left.” Please also edit figure 2 labels to be consistent with figure 3 (e.g. Bietry 2016 (Current) to Bietry 2016 (Current user))
3. The second argument to not including a subgroup odds ratio calculation is a little hard to follow. In line 194, the authors argued “dividing statin-users into current and former users itself suffers from potential lead-time bias as the definition of cholelithiasis in the included studies […] could not reflect the precise moment of the formation of gallstones…” and later in line 198 “It can take up to 20 years for gallstones to become symptomatic hampering the definition of current and former users.” In Table 1, current and former users are defined by the last prescription of statin, so it is unclear to me how the diagnosis of gallstone will hinder the definition of current and former users. Does the “lead-time bias” come from diagnosing the gallstone early for the former users? Or does it refer to the fact that former users are on statin for longer so they have a higher chance of also getting gallstone? It will clarify the argument if the authors could please rephrase some of the explanation.
4. In Figure 3, the odds ratio from former users in certain studies (e.g. Bietry 2016 and Bodmer 2009, to some degree Erichsen 2011) has higher variability and the odds ratio approaches no association when considering the higher end of the 95% CI. I understand the authors’ argument to not separately calculate odds ratio for the former and current patients due to arbitrary cutoff of description date, but could the authors please elaborate on this source of variability and whether or not it could be biologically significant (i.e. when the patients no longer take statin, the protective effect reduces?)
5. There are also a few typos and grammatical mistakes: e.g. in line 35 double bracket “((“ should be single bracket. Line 71: please consider changing “available and feasible literature” to “available and relevant literature.”

Reviewer 2 ·

Basic reporting

The authors need to improve their text in certain areas to communicate clear meaning.

Experimental design

The experimental design in thorough and legible.

Validity of the findings

Since the data have been extracted from already published manuscripts, the validity of findings are not in question.

Additional comments

The authors have performed a meta-analysis in which they aimed to find an association between the duration of use of statins and risk of gallstone disease. They were successful in analyzing the literature and supporting their conclusions. Overall it is a well thought out manuscript.
However, certain parts of the text can be improved for better understanding of the reader. For example,
in line 29, duration should be plural otherwise it just means a different time and not a collection of different times. This change should be applied everywhere in the text.
Line 47 should be completely rephrased. In its current form it fails to convey the meaning that statin users that have used statins long-term are at a lower risk of gallstone disease.
In line 52, authors provide a range of percentages therefore they should use between instead of around.
In line 59, additional should be addition.
In line 71, update should be replaced with determine because this is something that is not yet established.
In line 163, the authors need to add more information if they are using 'were' (past tense). For example they can add the year or duration when this treatment was used. If this is not the intent then the sentence should be changed to present tense.
Throughout the text, et al., should be presented as shown.
In line 230, add 'with' between associated and gallstone.

Reviewer 3 ·

Basic reporting

no comment

Experimental design

The abstract lists the main goal as the study of statin treatment duration and its association with gallstone disease. However, in the methods section, the main comparison is listed as statins users vs. non-statins users, and the study of temporal association is listed as an additional subgroup analysis in which the duration of treatment is treated as a confounder. The methods should be clarified as the current objective is not clear here.

Validity of the findings

The findings are somehow redundant. Indeed, for example, a manuscript from 2011 (10.1093/aje/kwq361; cited in the current manuscript and included in the meta-analysis) presents almost the same findings. Specifically, a section of the abstract from the 2011 manuscript lists “1.24 (95% CI: 1.11, 1.39), 0.97 (95% CI: 0.86, 1.10), and 0.79 (95% CI: 0.64, 0.97)” as odd ratios for short-term, medium-term, and long-term users, in the risk of gallstone disease. The statement that the impact of statin duration on gallstone disease has not been clarified might thus seem misleading. It might be more appropriate to state that the differences observed in odd ratios for treated vs. untreated observed across different studies can be imputed to the duration of treatment not being accounted for.

Additional comments

The following section of the introduction should be reworked as it might be considered misleading in its current form:
In the introduction, starting line 65 results from a meta-analysis from 2015 (Kan et al.) are compared to those of two other studies (Bietry et al. 2016 and Martin et al. 2015). The authors state that the results from the 2015 meta-analysis are different from those of the two other studies. However, this is not correct and additionally should be nuanced. Indeed, Bietry and colleagues found that long-term users had decreased odd ratios. Furthermore, Martin and colleagues found no significant differences in treated vs. untreated patients, but a similar trend. This should be nuanced with the fact that users were defined as 90 days of statin use or more, whereas Bietry and colleagues used 180 days as a threshold, which might explain why no significant association was uncovered by Martin and colleagues. This is overall consistent across cited studies.

There are some typos throughout the manuscript, as well as some incorrectly phrased sections including, but not limited to, the following:
1. Line 57: “due to its” should read as “due to their”.
2. On line 99, “maybe” should read “may be”.
3. In Figure 1, 942 records are excluded after duplicate removal. However, in the text, 958 records are listed are removed. I would advise the authors to check those numbers carefully.

Some additional comments that the authors might consider incorporating:
1. From line 65, it appears that citation numbering was not updated. It jumps directly from 14 to 22.
2. Rather than stating that databases were searched from “the inception” a date should be provided, as usually only studies up to a certain date are listed.
3. To confirm that combining RR and OR did not have an impact on overall results, it might be relevant to conduct two separate analyses, one for each type of ratio.
4. In the discussion, starting on line 171, the authors suggest that differences in practices are related to differences in accessibility and costs, however, this is not only irrelevant but no evidence is provided. I suggest the authors rephrase that section.
5. Differences in demographics, as reported in supplemental table 1, should be discussed in more detail.

---

## Round 0.2 · Minor Revisions

While most of the issues raised by the reviewers have been addressed by the authors, one of the reviewer has re-raised the concerns on some of the interpretations made in the manuscript. Please address the comments of reviewer #3.

Reviewer 1 ·

Basic reporting

no comment -- the authors have corrected the typos and clarified my questions

Experimental design

no comment

Validity of the findings

no comment

Reviewer 2 ·

Basic reporting

The authors have successfully addressed all the comments previously provided.

Experimental design

No comment

Validity of the findings

No comment

Reviewer 3 ·

Basic reporting

Regarding comment 2, I understand that there are variations across the studies for the association of statins use with gallstone disease, and that is, in essence, what the authors attempt to clarify. However, the main conclusion of the paper mentioned in the response to reviewers, seems to disagree with the authors’ interpretation. Indeed, the Bietry paper reports that, when considering, cholecystectomy, “Long-term current statin use (5–19 prescriptions) was associated with a reduced OR (aOR 0.77, 95% CI: 0.65, 0.92).”. Are you referring to a different paper?
In regards to this comment, the revised text “Biétry et al. reported comparable risk of gallstone disease among long-term user and nonusers.” might need to be adapted as well.

In one of the updated sections, there is a typo that the authors might have missed “For example, there [were?] wide variation in prevalence of diabetes, ischemic heart diseases, and cerebrovascular diseases among included studies.”

Experimental design

no comment

Validity of the findings

no comment

Additional comments

I believe the authors are still exaggerating their findings. Indeed, the meta-analysis does confirm the results of previous analyses and provide some additional insights, however, there are no novelties and this should be acknowledged more clearly in the manuscript.

---

## Round 0.3 · accepted · Accept

The authors have successfully addressed most of the comments and concerns of the reviewers and.I am happy to recommend this manuscript for publication.

Reviewer 3 ·

Basic reporting

no comment

Experimental design

no comment

Validity of the findings

no comment

Additional comments

no comment